# Effective Attention-Based Mechanism for Masked Face Recognition

**Vandet Pann and Hyo Jong Lee ***

Division of Computer Science and Engineering, CAIIT, Jeonbuk National University, Jeonju 54896, Korea;
pvd.vandet@jbnu.ac.kr
* Correspondence: hlee@jbnu.ac.kr

**Abstract:** Research on facial recognition has recently been flourishing, which has led to the introduction of many robust methods. However, since the worldwide outbreak of COVID-19, people have had to regularly wear facial masks, thus making existing face recognition methods less reliable. Although normal face recognition methods are nearly complete, masked face recognition (MFR)—which refers to recognizing the identity of an individual when people wear a facial mask—remains the most challenging topic in this area. To overcome the difficulties involved in MFR, a novel deep learning method based on the convolutional block attention module (CBAM) and angular margin ArcFace loss is proposed. In the method, CBAM is integrated with convolutional neural networks (CNNs) to extract the input image feature maps, particularly of the region around the eyes. Meanwhile, ArcFace is used as a training loss function to optimize the feature embedding and enhance the discriminative feature for MFR. Because of the insufficient availability of masked face images for model training, this study used the data augmentation method to generate masked face images from a common face recognition dataset. The proposed method was evaluated using the well-known masked image version of LFW, AgeDB-30, CFP-FP, and real mask image MFR2 verification datasets. A variety of experiments confirmed that the proposed method offers improvements for MFR compared to the current state-of-the-art methods.

**Keywords:** facial recognition; convolutional neural network; deep learning; masked face recognition; attention module

## 1. Introduction

Face recognition (FR) has represented one of the most important research topics for many years. Many researchers [1–6] have introduced robust methods to solve the FR problem. The trend of developing methods for FR appears to have almost reached its peak at the time of this writing. Influenced by the convolutional neural networks (CNNs), the current algorithms using deep learning methods [1–6] have achieved superior accuracy for FR. Systems based on FR are widely used in many areas across the world including airports, community gates, and healthcare; FR is also employed in some authentication applications, such as face-to-face attendance monitoring and mobile payment systems based on face profiles.

With the emergence of the COVID-19 pandemic, a viral infection caused by severe acute respiratory syndrome [7] has spread globally and brought many major challenges to daily human activities. To avoid COVID-19 infection, many people have worn and continue to wear masks. Mask wearing affects current FR application systems because the human face—the target of interest—is partially covered. In real-world FR applications, face occlusion, particularly masked face occlusion, will significantly affect existing FR performance and decrease re-identification accuracy [8].

Modern deep learning-based models are advanced enough to extract face features and learn the important key features such as face edges, mouth, nose, and eyes [9]. However,

the presence of a facial mask occludes most of the key features, thus complicating the feature extraction process. Since traditional methods for FR have been designed specifically to work with all face information available, a mask on a face makes the models lose about 50% of the useful information [10]. The facial mask blocks important features such as the mouth and nose, thus obstructing the face feature structure, as reported in [11]. This specific issue has recently emerged as a particularly serious barrier to the field of FR. Therefore, to solve this problem, novel methods must be invented, or the existing algorithms must be modified substantially.

Initially, facial mask recognition has been attempted, and researchers have introduced many robust solutions [12–14] to detect facial masks. Many scholars have recently presented various methods that address the MFR problems using deep learning techniques [9,11,15–20]. Alzu'bi et al. summarized the various MFR methods that have recently been proposed [21]. Further, because of the insufficient availability of masked face images for model training and testing, studies have proposed several masked face datasets [11,22,23] and data augmentation tools for generating simulated masked images [9,11].

MFR represents a special case in the occlusion FR domain. In contrast to regular occluded FR, MFR involves three major challenges: the key features of the face, such as the mouth, nose, and chin, are occluded; most of the FR methods have been designed specifically to work with all face features available, and there is currently no publicly available large-scale masked face training and testing benchmark dataset. Moreover, most existing methods have been developed for the specific masked face datasets used in their development. Thus, a specific method may perform well for a limited dataset whereas it performs badly for other datasets. Further, the average accuracy of the existing MFR methods is only 89.5% [21]. With this background, it is necessary to develop a method that can consistently achieve good results on all datasets.

Recently, the methods based on attention mechanism are widely used to solve various problems in vision tasks such as image classification [24], age-invariant face recognition [25,26], and specifically face recognition with masked face [11,19]. It should also be noted that the existing MFR methods which are based on attention demonstrate high accuracy compared to other methods. Hence, this paper proposes a method for solving the problem of MFR by verifying individuals with masked faces using an attention module and angular margin loss ArcFace. This method uses a refined ResNet-50 [5] network as a backbone network and integrates the attention module into the backbone network. The model can obtain highly discriminative features and improve facial feature representations through the proposed method, which successfully overcomes the recognition accuracy problem of MFR. However, recognizing the face of a person wearing a facial mask with hat, glass, and different face angles is a limitation of this approach.

The main contributions are summarized as four-fold:

- A new MFR method using a deep learning network architecture based on the attention module and angular margin loss ArcFace is proposed to focus on the informative parts not occluded by the facial mask (i.e., the regions around the eyes).
- The CBAM attention module is integrated with a refined ResNet-50 network architecture for feature extraction without additional computational cost.
- Proposed new simulated masked face images generated from regular face recognition datasets using a data argumentation tool for model training and valuation. Datasets generated in this research are available through the website https://github.com/MaskedFaceDataSet/SimulatedMaskedFaceDataset (accessed on 6 May 2022).
- The experimental results on simulated and real masked face datasets demonstrate that the proposed method outperforms other state-of-the-art methods for all datasets.

## 2. Related Works

With the success of FR research, researchers have continued to focus on the challenges posed by occluded face recognition [17,27,28]. The recognition of an occluded face is challenging because the human face can be covered by visual obstacles of any size or shape

appearing anywhere [29]. With the COVID-19 pandemic, MFR has become one of the greatest challenges in the FR domain. MFR is a specific facial occlusion problem since the essential parts of the face, such as the mouth, nose, or chin, are occluded. The objective of research on MFR is to identify or verify the specific identity of a person when people are wearing a facial mask. Some of the existing methods that researchers have proposed to solve occluded face recognition and MFR problems are described in this section.

Song et al. [17] presented a technique to address partial occlusion by discovering and disposing of corrupted face feature elements for recognition. This study decomposed the face recognition challenge under random partial occlusions in three stages: First, they developed a pairwise differential Siamese Network (PDSN) to capture the differentiation in the face features between the occluded and non-occluded face pairs. Second, they built a masked dictionary for masked features that they obtained from the previous stage to composite the feature discarding mask (FDM). Third, a combination of the FDM of random partial occlusions from the dictionary is multiplied by the original feature to eliminate the effect of partial occlusions from recognition. This approach aims to remove the occluded areas from depth features. However, it is difficult in practice to meet the requirements of the matched image.

Various studies have adopted restoration-based methods [30–33] to restore the missing part of the face image and reconstruct a new face image from the training dataset. Since generative adversarial nets (GANs) were first introduced [34], many researchers have used GAN methods to address facial occlusion problems. Yeh et al. [35] proposed a method that involved generating the corrupted pixel(s) and reconstructing the missing content. Din et al. [18] proposed a model that can detect and remove the mask to provide complete, unobstructed facial images. First, the model detects the mask region and produces it as binary segmentation. Then, it uses two discriminators based on the GAN network to learn the global structure and missing part of the facial image. However, these approaches have not evaluated the recognition performance of their models. In contrast to the previous GAN-based methods, Li et al. [36] presented an algorithm framework that consists of de-occlusion and distillation modules. The de-occlusion module uses GAN to perform masked face completion, which recovers the occluded features beneath the mask and eliminates the appearance uncertainty. The distillation module uses a pre-trained model to perform face classification. On the simulated LFW dataset, their highest accuracy for recognition performance is 95.44%.

MFR became an urgent research topic to consider during the COVID-19 epidemic. Mandal et al. [15] proposed a new framework with which to handle the MFR problem that used a deep network based on ResNet-50 [37]. The authors trained the network using the small Real-world Masked Face Recognition Dataset (RMFRD) described in [22]. However, this method did not yield adequate results because the network used only works with non-occlusion faces. Anwar and Raychowdhury [9] presented a similar strategy using FaceNet [1], a deep network-based face recognition system, to train with their dataset VGGFace2-mini-SM1. They used their own proposed simulated masked face dataset to train the network. This method produced better results than the first method since they trained with a large dataset from scratch.

Meanwhile, Huang et al. [38] used ArcFace [5], a deep network-based face recognition system, to train with their simulated dataset. Their simulated dataset was generated with random occlusion (mask or glasses). In that study, the network was able to learn more features than the masked dataset. However, their performance results greatly decreased when tested with only the masked face dataset. Walid Hariri [16] proposed a new method based on occlusion face removal and deep learning-based features to discard the occlusion region. They used a pre-trained network to handle the MFR problem. They applied the cropping filter technique to remove the occluded part covered by a facial mask and therefore extract only features in the non-masked face region. The occlusion removal technique can discard non-masked face areas from each image. However, it cannot guarantee a clean elimination of non-masked face parts since facial masks are not all placed in the same

position on the face. Moreover, their recognition performance results with both simulated masked face and real masked face images still need to be improved.

Recent works have attempted to deal with MFR using attention mechanisms. Li et al. [20] proposed a new strategy by integrating a cropping-based and attention-based approach with the CBAM [26]. The cropping-based process removes the masked face region from face images. They examined several cropping proportion cases of the input image to find the one that achieved the best recognition accuracy. In the attention-based process, the masked face features and features around the eyes were respectively given low and high weights. The authors reported that their approach achieved 92.61% MFR accuracy. In another study, Deng et al. [11] proposed an algorithm using cosine loss (MF-Cosface) to handle the MFR. As a result, their method improved the accuracy of masked face recognition compared to the first method based on attention. They also designed an Attention-Inception module that combined the CBAM with Inception-ResNet to help the model pay greater attention to the region not covered by the mask. This technique achieved a slight improvement in the verification task.

The existing works described above inspire our present work. By observing the strength of the attention module, which plays an important role in MFR work, this study extends them further by proposing a novel network architecture by integrating the attention module into the refined ResNet-50 network implemented in the ArcFace repository.

## 3. Proposed Method

### 3.1. Feature Extraction Network

Feature extraction—which is a crucial process in masked face recognition—aims to extract the key face components such as the eyes, nose, mouth, and texture from a face image. However, this process becomes more complicated when there is a mask covering the face in question. Therefore, the selection of the feature extracting network is a critical decision. The refined CNN architecture ResNet-50 implemented in ArcFace work is selected as a network backbone to extract the face features. This study follows [5] to modify the layer block in the third stage from the original ResNet-50 [37] architecture {3, 4, 6, 3} to {3, 4, 14, 3} layer blocks. Further, the improvement residual unit architecture is also applied to the network, which has a BN-Conv-BN-PReLu-Conv-BN structure and sets the stride as two for the second convolutional layer instead of the first one (as shown in Figure 1). After the last layer, the batch normalization, dropout, fully connected layer, and batch normalization (BN-Dropout-FC-BN) structure is used to obtain the final 512-D face embedding feature.

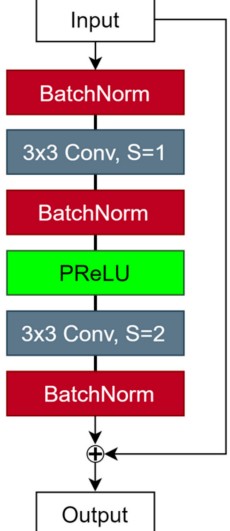

**Figure 1.** Structure of the improved residual unit: BN-Conv-BN-PReLu-Conv-BN.

### 3.2. Convolutional Block Attention Module (CBAM)

The proposed method adopts the CBAM presented by Woo et al. [24] in the network model. The CBAM consists of a channel attention module and a spatial attention module, which are arranged in a particular order, as shown in Figure 2. It is a lightweight module that can smoothly integrate with any CNN architecture. Given an input feature map $F \epsilon \mathbb{R}^{C \times H \times W}$ of the convolutional layer, where $C$, $H$, and $W$ are channel size, height, and width, respectively, let $M_{channel} \epsilon \mathbb{R}^{C \times 1 \times 1}$ denote a 1D channel attention map and $M_{spatial} \epsilon \mathbb{R}^{1 \times H \times W}$ denotes a 2D spatial attention map. The overall attention process can then be shown as shown in Equations (1) and (2).

$$F' = M_{channel}(F) \otimes F \tag{1}$$

$$F'' = M_{spatial}\left(F'\right) \otimes F' \tag{2}$$

where $\otimes$ denotes element-wise multiplication and $F''$ is the final output of the feature maps or refined feature maps.

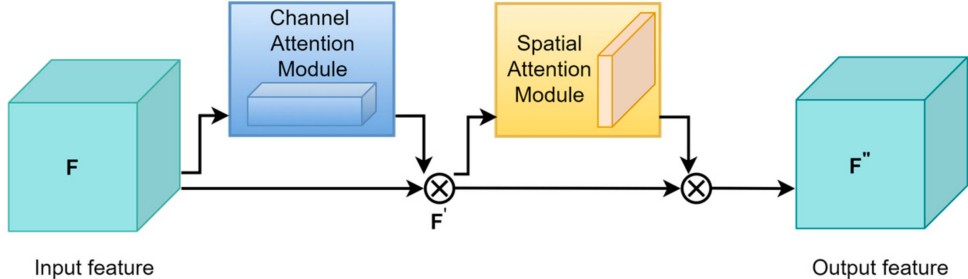

**Figure 2.** Overall structure of the convolutional block attention module.

1. Channel Attention Module

The channel attention module focuses on the major features of the input image. This module uses both average-pooling and max-pooling operations on the input feature map to generate two different spatial information vectors: $F_{avg}^{ch}$ and $F_{max}^{ch}$, which denote average-pooled features and max-pooled features, respectively. Both vectors are consecutively forwarded to a shared network multi-layer perceptron (MLP) with filter kernel size $1 \times 1$ to produce a channel attention map $M_{channel} \epsilon \mathbb{R}^{C \times 1 \times 1}$. Next, the output feature vectors from the shared network are merged using element-wise submission. The final output of the $M_{channel} \epsilon \mathbb{R}^{C \times 1 \times 1}$ after element-wise submission is then passed to the sigmoid function $\sigma$ to generate the channel weights, as shown in Equation (3). The channel attention module process can be depicted as shown in Figure 3.

$$M_{channel}(F) = \sigma(MLP(AvgPooling(F)) + MLP(MaxPooling(F))) \tag{3}$$

where $\sigma$ is the sigmoid function and $MLP$ uses the ReLu activation function.

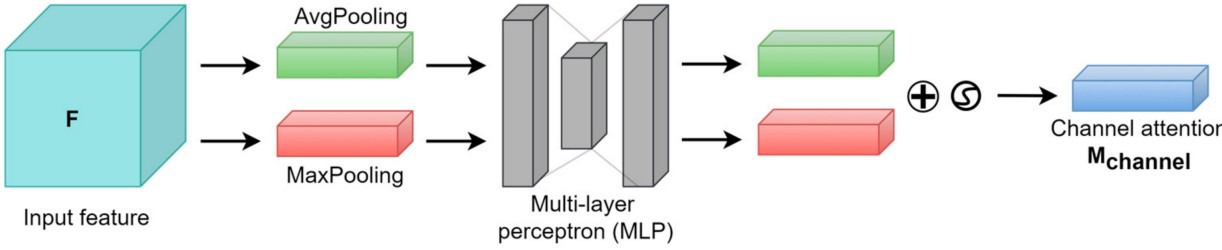

**Figure 3.** Structure of the channel attention module.

2. Spatial Attention Module

The spatial attention module focuses on an informative region of the input images features. Similar to the channel attention module, the spatial attention module adopts the average-pooling and max-pooling operations to obtain two 2D maps: $F_{avg}^{sp}$ and $F_{max}^{sp}$ denote average-pooling and max-pooling features, respectively. Those are then concatenated with a convolution layer with a filter kernel size of $7 \times 7$ to obtain a 2D spatial attention map $M_{spatial} \in \mathbb{R}^{1 \times H \times W}$. The spatial attention module process can be illustrated as shown in Figure 4 and Equations (4) and (5).

$$M_{spatial}(F) = \sigma\left(f^{7 \times 7}([AvgPooling(F); MaxPooling(F)])\right) \tag{4}$$

$$= \sigma\left(f^{7 \times 7}\left(\left[F_{avg}^{sp}; F_{max}^{sp}\right]\right)\right) \tag{5}$$

where $\sigma$ is the sigmoid function and $f^{7 \times 7}$ denotes a convolution operation with the filter kernel size of seven.

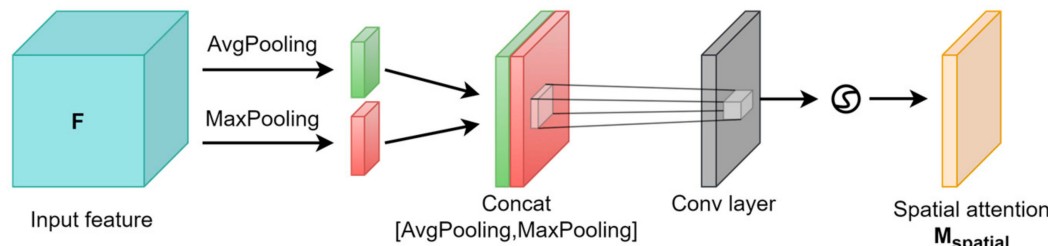

**Figure 4.** Structure of the spatial attention module.

*3.3. Network Architecture*

Figure 5 shows the overall proposed network architecture diagram. As described in Section 3.1, this work uses the refined ResNet-50 architecture as a backbone to extract face features. The proposed network model uses no-masked and masked face images with the size $3 \times 112 \times 112$ as the input. The network backbone architecture consists of four main convolutional layer block stages with the number of blocks stacked. Therefore, the respective numbers of blocks stacked in the first, second, third, and fourth stages are {3, 4, 14, 3}. The sizes of the feature maps in the first, second, third, and fourth stages are $64 \times 56 \times 56$, $128 \times 28 \times 28$, $256 \times 14 \times 14$, and $512 \times 7 \times 7$ with kernel size of $3 \times 3$, respectively. CBAM is adopted in each output of the convolutional block of the backbone network to focus more effectively on an object of interest effectively. *F* represents the feature map after the pre-operation of the convolution. Then, the channel and spatial attention modules compute sequentially to produce refined feature maps $F''$. Finally, the refined output features $F''$ are summed with the input feature maps of the previous block. The network repeats the same operation until the last convolutional layer block and the batch normalization (BN), dropout, and fully connected layers are applied to obtain 512-D face embedding features. ArcFace loss adds an angular margin *m* to the target (ground truth) and multiplies by the feature scale *s*. Then, the softmax function proceeds and contributes to the cross-entropy loss. This technique helps optimize the embedding feature to obtain highly discriminative features for MFR.

*3.4. Loss Function*

The loss function helps optimize the model and stabilize the training process. This method uses Additive Angular Margin Loss (ArcFace) [5], a margin loss function constructed by modifying the softmax loss function, which improves the discriminative power of the model. Furthermore, ArcFace optimizes the feature embedding to have the smallest

distance possible among the same classes and the largest distance possible among the different classes. ArcFace can be defined as follows:

$$L_{ArcFace} = -\frac{1}{N} \sum_{i=1}^{N} log \frac{e^{s(\cos(\theta_{y_i}+m))}}{e^{s(\cos(\theta_{y_i}+m))} + \sum_{j=1, \, j \neq y_i}^{n} e^{s \, \cos \theta_j}} \tag{6}$$

where $\theta_j$ denotes the angle between the weight and deep features, $s$ denotes the feature scale, $m$ denotes angular margin penalty, and $N$ and $n$ respectively denote batch size and class number.

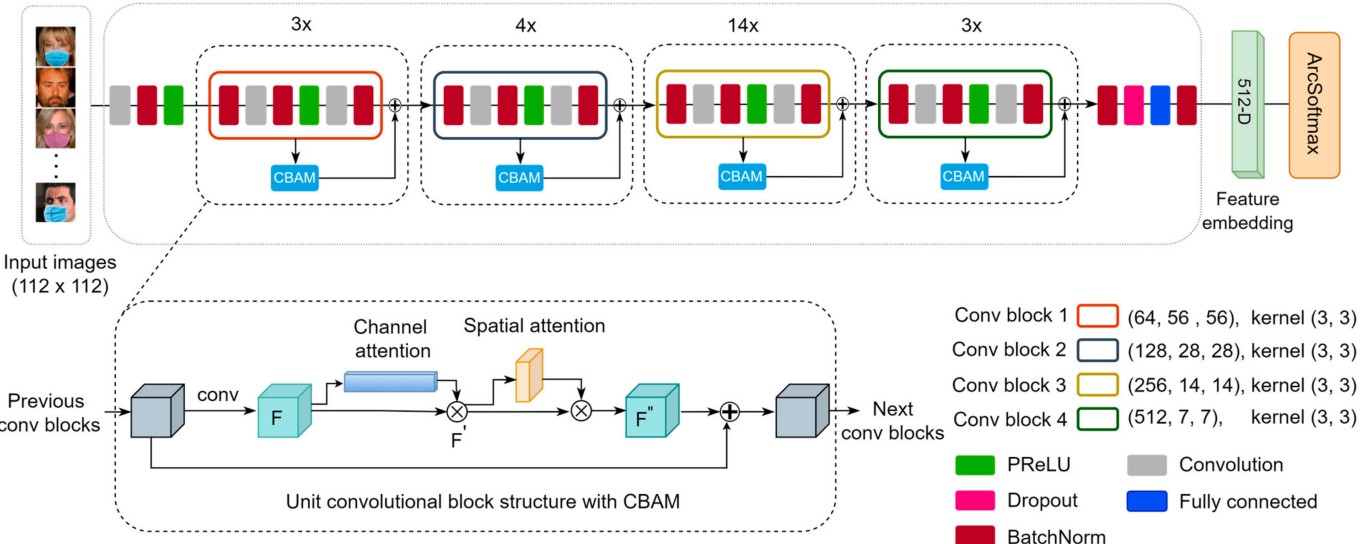

**Figure 5.** Overall structure of the proposed network architecture. The convolutional block attention module (CBAM) is integrated into each output of the block. Input image includes masked face and non-masked images of size $112 \times 112$.

## 4. Experiments and Results

### 4.1. Datasets

The developed network needs to be verified on both simulated and real masked face datasets. A data augmentation method presented by [9] is used to generate the masked face images version e from the existing normal face datasets for model training and evaluation. First, a multi-task cascaded convolutional neural network (MTCNN) [39] is used to detect faces from the raw images. The MTCNN detects the face and obtains five facial landmark key points: nose, right-eye, left-eye, right-mouth, left-mouth, and then face alignment and rotation are performed. To generate more realistic masked face images the method uses Dlib [40] library to detect 68 key points of the face. Lastly, to overlay a mask on the face, the method calculates the masked positions of the face and selects the suitable facial mask. All generated masked face datasets are listed in Table 1. A small set of real masked face MFR2 [9] is also used to evaluate the model.

**Table 1.** Summary of datasets used for model training and evaluation.

| Dataset | Type | Identities | Images |
|---|---|---|---|
| CASIA-WebFace_m | Simulated mask | 10,575 | 789,296 |
| LFW_m | Simulated mask | 5749 | 12,000 |
| AgeDB-30_m | Simulated mask | 568 | 12,000 |
| CFP-FP_m | Simulated mask | 500 | 14,000 |
| MFR2 | Real masked faces | 53 | 269 |

CASIA-WebFace_m is generated from CASIA-WebFace [41] dataset for model training. This dataset is a large-scale public face recognition dataset. It contains 494,414 images of 10,575 unique identities. During masked face generation, around 20% of face images could not be detected by the data augment tool. Therefore, after masked face generation, 394,648 masked images remain. The generated masked face image version is then combined with the corresponding regular face images to produce CASIA-WebFace_m for the model training. This means that the total training samples are 789,296 images.

More masked face images are generated from the most widely used benchmark dataset, LFW [42], AgeDB [43], and CFP [44], respectively. MFR2 [9] is a genuine masked face dataset instead of a simulated mask dataset. LFW_m, AgeDB-30_m, CFP-FP_m, and MFR2 datasets among them are used for model evaluation. Each simulated dataset is described briefly here.

- LFW_m is generated from the LFW dataset, which is most used for face verification. This dataset contains 5749 unique identities and a total of 13,233 face images. The experiment in this paper follows the LFW standard protocol using 6000 predefined comparison pairs, of which 3000 pairs have the same identities and the other 3000 pairs have different identities.
- AgeDB-30_m is generated from the public benchmark dataset AgeDB, which is an unconstrained face recognition dataset which is most used for cross-age face verification. This dataset contains 568 unique identities and a total of 16,588 face images. The experiment follows the protocol of AgeDB-30 using 6000 predefined comparison pairs, of which 3000 pairs have the same identities and the other 3000 pairs have different identities.
- CFP-FP_m is generated from the public benchmark dataset CFP, which contains 500 celebrities in frontal and profile views. This dataset has two verification protocols: CFP-FF and CFP-FP. In the experiment, the method uses the CFP-FP protocol using 7000 predefined comparison pairs, of which 3500 pairs have the same identities and the other 3500 pairs have different identities.
- MFR2 is a small set of real masked face images. It contains 53 identities of celebrities and politicians among 269 images, where each identity has an average of five images. This dataset consists of strange mask patterns. We collect 800 pairs of images for real masked face verification in the experiment. This means that 400 pairs have the same identities whereas 400 pairs have different identities.

Typical sample images of different datasets are shown in Figure 6.

### 4.2. Experimental Setting

Initially, this work follows [5] to generate normalized face crops ($112 \times 112$) in the data processing and applies the Batch-Normalization (BN) [45]-Dropout [46] structure after the last convolutional layer to obtain the output embedding feature of 512D. Dropout can effectively help avoid over-fitting and obtain a better generalization for deep face recognition. In the experiment, the dropout parameter is set to 0.4. The feature scale $s$ is set as 64 based on [4] and angular margin penalty $m$ is chosen as 0.5 based on [5]. All experiments in this work are implemented using Python programing language and a Pytorch-based [47] open-source deep learning framework. The batch size is 128, and the model is trained on NVIDIA Quadro RTX 6000 (48GB) GPUs. The overall model architecture is trained up to 100 epochs, and the only CASIA-WebFace_m dataset is used to train the model. The learning rate was set as 0.01 and divided by ten at 13 and 21 epochs. Lastly, the momentum and weight decay are set as 0.9, and $5 \times 10^{-4}$, respectively.

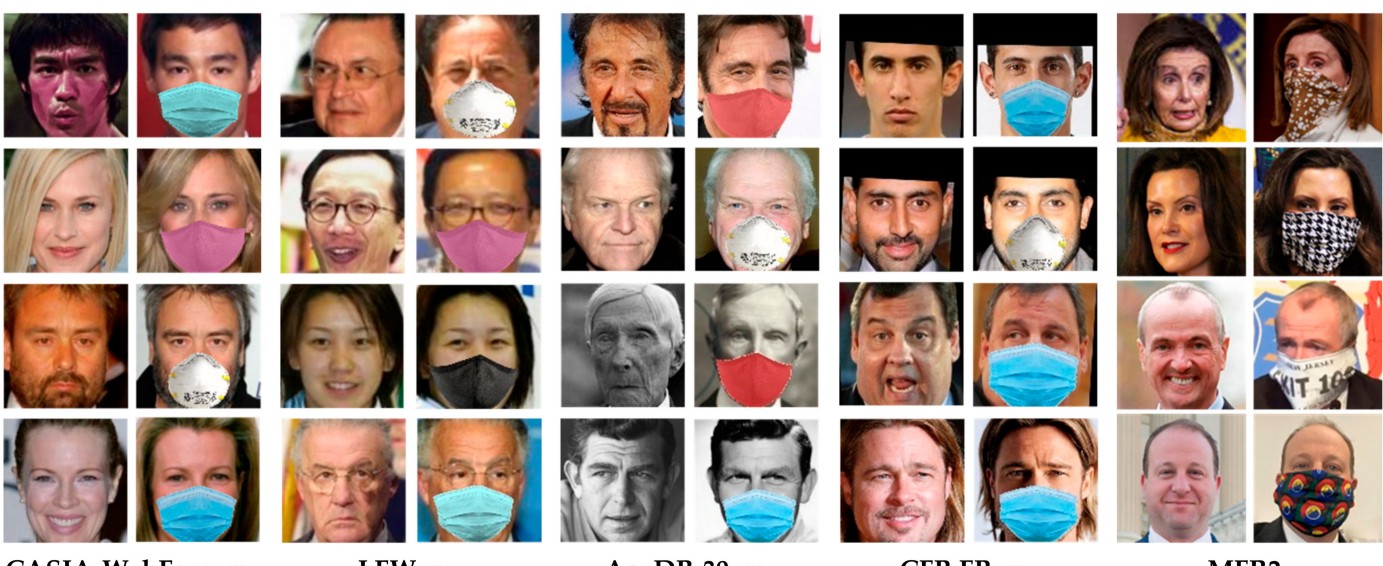

| CASIA-WebFace_m | LFW_m | AgeDB-30_m | CFP-FP_m | MFR2 |

**Figure 6.** Sample images of training and validation dataset. CASIA-WebFace is a training dataset whereas the rest are validation datasets. Each dataset has two columns of corresponding images.

### 4.3. Evaluation Metrics

To assess the proposed method, four evaluation parameters, accuracy, precision, recall and F1 score are adopted.

**Accuracy.** The accuracy is an intuitive performance measure, and it is defined to describe the accuracy of the algorithm for recognition and classification problems. It represents the ratio of the correctly predicted sample to the total of sample, which can be computed as shown in Equation (7).

$$Accuracy = (TP + TN)/(TP + TN + FP + FN), \tag{7}$$

where TP, TN, FP, and FN are true positive, true negative, false positive, and false negative, respectively.

**Precision.** The precision is a metric that determines the number of accurate positive predictions. Therefore, precision computes the accuracy for the minority class. It is computed as the ratio of correctly predicted positive samples divided by the predicted number of positive samples. Precision can be computed as defined in Equation (8).

$$Precision = TP/(TP + FP) \tag{8}$$

**Recall.** The recall is a metric that measures the number of correct positive predictions made from all positive predictions that could have been made. This is as opposed to precision, which only considers the correct positive predictions out of all positive predictions. Recall can be computed as defined in Equation (9).

$$Recall = TP/(TP + FN) \tag{9}$$

**F1 score.** The F1 score allows for precisions and recalls to be combined into a single measure that captures both properties. It can express high precision with poor recall or, alternately, terrible precision with perfect recall. The F1 score can be computed as defined in Equation (10).

$$F1 = 2 \times (Precision \times Recall)/(Precision + Recall) \tag{10}$$

### 4.4. Experimental Results

This section reports the model evaluation results. We performed experiments in the face verification task and used the 10-fold cross-validation technique to evaluate the predictive model by randomly dividing the evaluation dataset into ten partitions: nine partitions are used as a training set whereas the remaining partition is used as a validation set. The model repeated training ten times and used the average of the ten validation results as the recognition accuracy. The model was evaluated on simulated masked face images LFW_m, AgeDB-30_m, CFP-FP_m, and real masked face images MFR2. The model extracts the features of all face pairings and then computes the cosine similarities between the face pairs. The accuracy is expressed as the percentage of right predictions, with the highest accuracy being chosen as the threshold. Table 2 reports measurements of the performance of the model in terms of accuracy, precision, recall, and F1 score metrics. The results show that the proposed method achieved high performance in the face verification task. The average accuracy of 10-fold cross-validation on the LFW_m, AgeDB-30_m, and CFP-FP datasets reached rates of 99.43%, 95.86%, and 97.74%, respectively. MFR2 achieved a rate of 96.75%, since this dataset contains different facial postures, expressions, and cloth masks in different textures and colors.

**Table 2.** Results of recognition accuracy, precision, recall, and F1 score (%) using CASIA-Webface_m.

| Dataset | Accuracy | Precision | Recall | F1 Score |
|---------|----------|-----------|--------|----------|
| LFW_m | 99.43 | 99.30 | 99.56 | 99.43 |
| AgeDB-30_m | 95.86 | 93.83 | 97.82 | 95.78 |
| CFP-FP_m | 97.74 | 96.77 | 98.69 | 97.72 |
| MFR2 | 96.75 | 96.25 | 97.22 | 96.73 |

We conducted experiments with other state-of-the-art FR methods. Only the proposed method used the CASIA-WebFace_m dataset, as other methods used the original CASIA-WebFace dataset from scratch. The results of the verification accuracies were compared by validating on the same validation dataset. The recognition accuracy results are listed in Table 3.

**Table 3.** Comparison of face verification results (%) on validation dataset with different methods.

| Method | Training Set | LFW_m | AgeDB-30_m | CFP-FP_m | MFR2 |
|--------|--------------|-------|------------|----------|------|
| CosFace [4] | CASIA-Webface | 95.23 | 93.40 | 92.21 | 63.00 |
| Softmax [5] | CASIA-Webface | 96.68 | 93.50 | 94.78 | 69.75 |
| ArcFace [5] | CASIA-Webface | 96.85 | 94.10 | 95.10 | 71.87 |
| Proposed method | CASIA-Webface_m | **99.43** | **95.86** | **97.74** | **96.75** |

As reported in Table 3, our method yielded better results in both generated masked face images and real masked face images. The accuracy rates with the generated images are high and comparable to the results of the existing FR methods. While the accuracy rates of compared methods drop considerably with real mask images (MFR2), the proposed method maintains similar accuracy throughout all benchmark datasets.

Several MFR methods are conducted with their proposed training and validation datasets. To compare the proposed method to current existing MFR methods, this study separated the comparison into two parts: In the first part, we compared the presented method results with the results of other MFR methods, as presented in Table 4. In the second part, another experiment was conducted to compare the current advanced method MFCosface [11] with their masked dataset VGG-Face2_m. We trained the proposed network model using the same VGG-Face2_m and tested with 400 pairs of the MFR2 dataset for face verification. The verification performance of the recognition accuracy, precision, recall, and F1 score results are shown in Table 5. Tables 4 and 5 show that the proposed method performs slightly better than MFCosface [11] for both LFW_m and MFR2 datasets, if it is

trained with VGGFace2_m. However, MFCosface shows better performance with MFR2 when the proposed method is trained with CASIA-Webface_m.

**Table 4.** Comparison of face verification results (%) with different methods.

| Method | Training Set | LFW_m | AgeDB-30_m | CFP-FP_m | MFR2 |
|--------|--------------|-------|------------|----------|------|
| Huang et al. [38] | WebFace-OCC | 97.08 | 87.18 | 86.07 | - |
| Anwar et al. [9] | VGGFace2-mini-SM | 97.25 | - | - | 95.99 |
| MFCosface [11] | VGG-Face2_m | 99.33 | - | - | **98.50** |
| Proposed method | CASIA-Webface_m | **99.43** | **95.86** | **97.74** | 96.75 |

**Table 5.** Results of accuracy, precision, recall, and F1 score (%) using VGG-Face2_m.

| Dataset | Accuracy | Precision | Recall | F1 Score |
|---------|----------|-----------|--------|----------|
| LFW_m | 99.41 | 99.26 | 99.56 | 99.40 |
| AgeDB-30_m | 95.38 | 93.10 | 98.11 | 95.53 |
| CFP-FP_m | 96.98 | 96.17 | 98.40 | 97.27 |
| MFR2 | 99.00 | 99.50 | 98.54 | 99.02 |

*4.5. Ablation Experiments*

To prove the effectiveness of the proposed method, ablation experiments were performed. All experimental settings—including image size, batch size, and learning rate were applied—to match the previous experiments. First, we experimented with the CBAM attention module on proposed masked face dataset, and then explored each attention module with the backbone. We searched for an effective approach to channel attention and then spatial attention using our backbone network. Then each of the experimental models was evaluated on all validation datasets. Table 6 shows the performant reports of the ablation experiments. It can clearly be seen that the best performance is achieved when both channel and spatial attention modules are applied throughout all datasets.

**Table 6.** Ablation experimental results (%).

| Method | LFW_m | AgeDB-30_m | CFP-FP_m | MFR2 |
|--------|-------|------------|----------|------|
| CBAM | 98.66 | 94.45 | 96.15 | 95.50 |
| Backbone | 99.31 | 95.28 | 97.08 | 96.25 |
| Backbone + $M_{channel}$ | 99.35 | 95.53 | 97.47 | 96.50 |
| Backbone + $M_{spatial}$ | 99.38 | 95.58 | 97.38 | **96.75** |
| Backbone + $M_{channel}$ + $M_{spatial}$ | **99.43** | **95.86** | **97.74** | 96.75 |

## 5. Discussion

MFR is a significantly challenging problem that is currently attracting substantial research interest in computer vision and the face recognition field. As the key features such as the mouth, nose, and chin are occluded by mask wearing, existing face recognition methods perform poorly. Further, the insufficient availability of training and validating datasets currently represents a major barrier to the adoption of deep learning approaches in MFR. Figure 7 illustrates the loss and accuracy curves of the model. The loss curve shows that the proposed model is learning from the data by trying to reach the minimum point and the accuracy curve still slightly increase until the last epoch. The experimental results show that the proposed method can achieve high performance in the verification task on simulated masked datasets. However, this method exhibited slightly decreased performance when evaluated on the real masked dataset due to the small size of the training real face data. By contrast, other methods exhibited a substantial decrease in performance when evaluated on the real masked dataset.

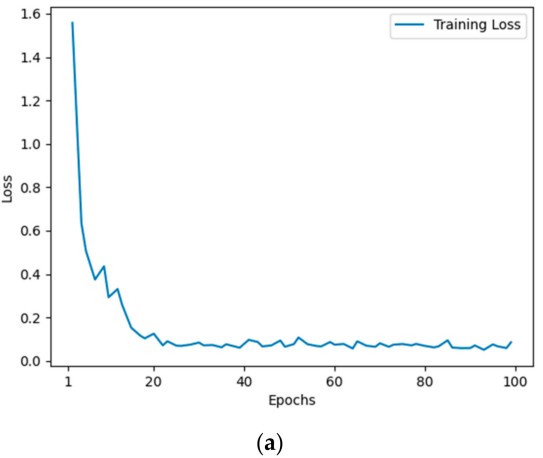

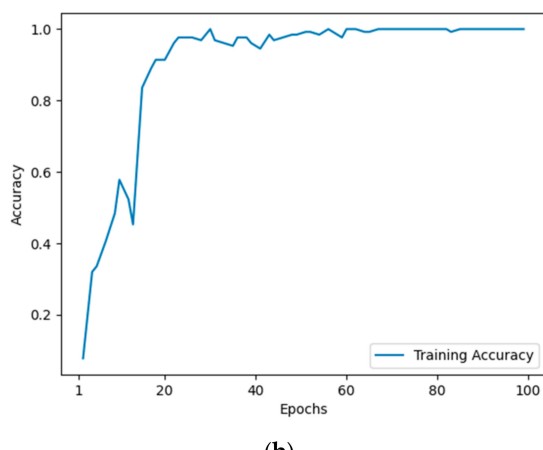

(**a**)     (**b**)

**Figure 7.** Training curves of the proposed method. (**a**) Training loss curve. (**b**) Training accuracy curve.

## 6. Conclusions

This paper presents a new method to solve the masked face recognition problems using deep learning technology. Traditional FR methods based on deep learning can address normal face recognition problems and achieve high performance. However, such methods show dramatically reduced performance when a face is covered with a mask. Through the analysis of the masked face images, we found that some of the key facial features are covered by a facial mask which makes the FR methods cannot recognize the face properly. To tackle the problem, this study introduced a new network architecture based on an attention mechanism that can focus on the most informative part around the eyes of the masked face images and obtain more discriminative feature information. Moreover, one of the most widely used ArcFace loss functions is implemented into the proposed network to optimize the feature embedding and to increase the similarity of the intra-class samples and diversity of the inter-class sample. To handle the problem of insufficient masked face datasets, new simulated masked face images are generated by using data augmentation for model training and evaluation. Through the various experiments, the following points summarize the findings in this paper:

- The attention module can focus on the non-occluded part of the masked face and significantly improve the recognition performance.
- The newly generated masked face dataset can effectively help the model training and evaluation.
- The results show that the proposed method provides outstanding performance and a better recognition rate on both generated masked face and real masked image datasets compared to the state-of-the-art methods.

We hope this research study becomes a useful solution to solve the masked face recognition problem. In future work, the improvement of the method to solve masked face recognition with different postures, expressions, illumination, and the presence of a hat are considered.

**Author Contributions:** V.P. designed and developed the proposed method, conducted the experiments and wrote the manuscript. H.J.L. designed the new concept, provided the conceptual idea and insightful suggestions to refine it further, and reviewed the manuscript. All authors have read and agreed to the published version of the manuscript.

**Funding:** This research was supported by Basic Science Research Program through the National Research Foundation of Korea (NRF), funded by the Ministry of Education (GR2019R1D1A3A03103736).

**Institutional Review Board Statement:** Not applicable.

**Informed Consent Statement:** Informed consent was obtained from all subjects involved in the study.

**Data Availability Statement:** The datasets are available at the following link: https://github.com/MaskedFaceDataSet/SimulatedMaskedFaceDataset (accessed on 6 May 2022).

**Conflicts of Interest:** The authors declare no conflict of interest.

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
