# Peer review of "Effective Attention-Based Mechanism for Masked Face Recognition"

_applsci, doi:10.3390/app12115590_

Round 1

Reviewer 1 Report

The manuscript proposes a variety of depth learning methods based on convolution block attention module (CBAM) and angle edge arc loss. In this model, CBAM is combined with convolutional neural network (CNNs) to extract the feature map of the input image, especially the feature map of the periocular region.

The method is evaluated by the famous mask image LFW, agedb-30, CFP FP and real mask image MFR2 validation data sets. Many experiments show that this method provides an improvement for MFR com-22, which is compared with the most advanced method at present.

  1. Introduction to the manuscript. In order to better guide readers to read the article, it is suggested to add some content to the introduction part of the manuscript and arrange the rest of the manuscript. Please refer the recent papers: Chenggang Yan, Biao Gong, Yuxuan Wei, Yue Gao, “Deep Multi-View Enhancement Hashing for Image Retrieval”, IEEE Transactions on Pattern Analysis and Machine Intelligence, 2020.Chenggang Yan, Zhisheng Li, Yongbing Zhang, Yutao Liu, Xiangyang Ji, Yongdong Zhang, “Depth image denoising using nuclear norm and learning graph model”, ACM Transactions on Multimedia Computing Communications and Applications 2020.Chenggang Yan, Yiming Hao, Liang Li, Jian Yin, Anan Liu, Zhendong Mao, Zhenyu Chen, Xingyu Gao, “Task-Adaptive Attention for Image Captioning”, IEEE Transactions on Circuits and Systems for Video Technology, 2021. Chenggang Yan, Tong Teng, Yutao Liu, Yongbing Zhang, Haoqian Wang, Xiangyang Ji, “Precise No-Reference Image Quality Evaluation Based on Distortion Identification”, ACM Transactions on Multimedia Computing Communications and Applications 2021.Chenggang Yan, Lixuan Meng, Liang Li, Jiehua Zhang, Jian Yin, Jiyong Zhang, Zhan Wang, Bolun Zheng, “Age-Invariant Face Recognition By Multi-Feature Fusion and Decomposition with Self-Attention”, ACM Transactions on Multimedia Computing Communications and Applications 2021
  2. About the picture annotation of the manuscript. In order to make readers better distinguish between the text and notes, it is suggested that the font of the notes part of the manuscript is different from that of the text part, which can remind readers and make the boundary of the article clearer.
  3. The manuscript proposes an MFR method based on deep learning network structure. The method focuses on the unobstructed information part of the mask (i.e. the area around the eyes) based on the attention module and the missing arc of the corner edge.
  4. The manuscript uses a data enhancement method to generate a new simulated mask face image from the conventional face recognition data set for model training and val evaluation. The data sets generated in this study can be https://github.com/MaskedFaceDataSet/SimulatedMaskedFaceDataset. Website access.

Reviewer 2 Report

Dear Authors

The manuscript is interesting due to the current global concerns and it has the potential to be considered for publication in this journal.

The language is good although some minor typos have been found. Please review all the text and double check the grammar and phrase.

I do recommend the authors to address the reviewer’s concerns in the revised version. Regarding the figures, they are well presented.

  1. The paper missed novelty and innovation concept! There are some similar works in the literature. Please stress the novelty of the work in the introduction section by emphasizing the advantage and limitation of this study and also compared to the already published works. Otherwise, it cannot be a novel research work.
  2. The manuscript lacks a table of nomenclature since it includes plenty of variables and acronyms.
  3. Please avoid using active tense starting with “WE”, “OUR” in the academic manuscript! In the revised version, it is highly recommended to use the passive tense.
  4. How did you arrive at Equation (3)? It is vague!
  5. Section 3.3, regarding the sizes of the feature maps; please clarify the intention to select these values.
  6. Have you made any type of programming? Matlab codes? Please comment it.
  7. Please reduce the subsections! Why did you define too many subsections in this short paper?
  8. About the masked face generation in section 4.1, more details are needed to clarify the concept. Please add it to the revised version.
  9. Section 4.2, “The feature scale ? and angular margin penalty ? are respectively set as 64 and 0.5”, how did you define these factors? Please comment on it.
  10. Table 3 and Table 4, what do you mean of last row “OURS”? It is better to state that “present method”, for instance.
  11. Can you please present deviation on the results obtained within your methodology? I cannot find this type of results in the paper.
  12. Conclusion is vague! I suggest to rewrite it and present it with some bullets stressing the main outcome of the work.

Very Best

The Reviewer

Round 2

Reviewer 2 Report

Dear Authors

Good improvement, at this stage I Recommend an acceptance!

Good luck

Best regards

The Reviewer